



# Observations of cold cloud properties in the Norwegian Arctic using ground-based and spaceborne lidar

Britta Schäfer[1], Tim Carlsen[1], Ingrid Hanssen[2], Michael Gausa[2], and Trude Storelvmo[1,3]

[1]Department of Geosciences, University of Oslo, Oslo, Norway
[2]Andøya Space Center, Bleiksveien 46, 8480 Andenes, Norway
[3]Nord University, Bodø, Norway

**Correspondence:** Britta Schäfer (britta.schafer@geo.uio.no)

**Abstract.** The role of clouds for the surface radiation budget is particularly complex in the rapidly changing Arctic. However, despite their importance, long-term observations of Arctic clouds are relatively sparse. Here we present observations of cold clouds based on 7 years (2011-2017) of ground-based lidar observations at the Arctic Lidar Observatory for Middle Atmosphere Research (ALOMAR) in Andenes in the Norwegian Arctic. In two case studies, we assess (1) the agreement between a
collocated cirrus cloud observation from the ground-based lidar and the spaceborne lidar onboard the Cloud-Aerosol Lidar and Infrared Pathfinder Satellite Observation (CALIPSO) satellite, and (2) the ground-based lidar's capability of determining cloud phase in mixed-phase clouds from depolarization measurements. We then compute multi-year statistics of cold clouds from both platforms with respect to their occurence, cloud top and base height, cloud top temperature and thermodynamic phase for the period 2011-2017. We find that satellite and ground-based observations agree well for the coincident cirrus measure-
ment, and that the vertical phase distribution within a liquid-topped mixed-phase cloud could be identified from depolarization measurements. On average, 8% of all satellite profiles were identified as single-layer cold clouds with no apparent seasonal differences. The average cloud top and base heights combining the ground-based and satellite instrument are 9.1 km and 6.9 km, respectively, resulting in an average thickness of 2.2 km. Seasonal differences between the average top and base heights are on the order of 1-2 km and are largest when comparing autumn (highest) and spring (lowest). However, seasonal variations are
small compared to the observed day-to-day variability. Cloud top temperatures agree well between both platforms with warmer cloud top temperatures in summer. The presented study demonstrates the capabilities of long-term cloud observations in the Norwegian Arctic from the ground-based lidar at Andenes.

## 1  Introduction

Clouds play an important role in the Earth´s radiative energy budget and hydrological cycle. While clouds cool the surface by
scattering incoming shortwave (SW) radiation back to space, they warm the surface by absorbing and emitting longwave (LW) radiation. The balance of these two processes determines the net effect of clouds on the surface radiation budget and is mainly determined by the cloud's macrophysical (e.g., occurence, cloud altitude, vertical extent, optical thickness) and microphysical (e.g., thermodynamic phase, water content, particle size and shape) properties. Due to their high altitude and low temperature, cirrus clouds generally have a warming effect on climate by reducing the emission of LW radiation to space, while low-level





clouds contribute to cooling by reflecting incoming SW radiation. This has been quantified by Matus and L'Ecuyer (2017) on a global scale using satellite observations. They highlight the crucial role of the cloud's thermodynamic phase composition for their radiative properties (e.g., Sun and Shine, 1994). The amount of liquid droplets and ice crystals in a cloud further controls the formation of precipitation and influences cloud lifetime (e.g., Korolev et al., 2017).

In a warming climate, cloud properties are expected to change, and, in turn, influence changes in the climate system through

feedback mechanisms. The latest report by the Intergovernmental Panel on Climate Change (IPCC) states that the net cloud feedback in a warming climate is positive, i.e. that changes in clouds amplify future warming (Forster et al., 2021). This is due to an increase in the altitude of tropical high clouds and a reduction in the occurrence of sub-tropical low-level clouds, while changes in extratropical clouds' composition from ice to more liquid water content have a counteracting, but weaker cooling effect. A focus region for studying clouds and cloud changes is the Arctic, because it is warming at a particularly high rate

compared to the global average, a phenomenon that is known as Arctic amplification (Serreze and Barry, 2011; Wendisch et al., 2017).

However, assessing how clouds influence the surface radiation budget is particularly complex in the high latitudes, where the dry atmosphere, high surface albedo due to snow and ice cover, lack of solar radiation in winter, and strong temperature inversions strongly influence the cloud's radiative effect (Curry et al., 1996). Intrieri et al. (2002) found that Arctic clouds

warm the surface for most of the year. However, for a brief period in summer they report a net cooling effect when the SW cooling outweighs the LW warming due to a lower surface albedo and larger solar elevations. While this has been observed in different regions of the Arctic, Miller et al. (2015) showed a continuous warming effect of clouds at Summit, Greenland, where the surface albedo is high throughout the year. The cloud radiative effect in the Arctic is dominated by clouds that contain liquid water (Shupe and Intrieri, 2004), and modeling studies suggest that the amount of liquid cloud water is essential

in understanding Arctic climate change (Hofer et al., 2017, 2019). However, Ebell et al. (2020) showed with ground-based remote sensing that cirrus clouds can dominate the LW radiative effect in the Arctic in winter.

Besides their radiative impact, Arctic cirrus clouds furthermore have the potential to dry the upper troposphere, contribute to chemical reactions affecting ozone, and redistribute trace gases and ice nucleating particles (INPs) which in turn affects lower mixed-phase clouds (Kärcher, 2005).

To estimate the radiative impact of Arctic clouds, long-term observations of their macrophysical and microphysical properties are needed (e.g., Turner et al., 2018). However, continuous cloud observations in the harsh and remote Arctic are scarce. The weak contrast between clouds and the underlying bright snow and ice surfaces makes passive remote sensing from satellite difficult to evaluate. Active radar and lidar measurements onboard the CloudSat (Stephens et al., 2002) and Cloud-Aerosol Lidar and Infrared Pathfinder Satellite Observation (CALIPSO) (Winker et al., 2009) satellites are valuable to provide cloud

observations in the Arctic, but their polar orbits limit their coverage to below 81°N and reduce the temporal resolution. Thus, for long-term observations of Arctic clouds, ground-based remote sensing sites are essential. Shupe et al. (2011) combined observations from six different Arctic sites and estimated the total annual cloud occurence to 58–83%, whereas cloud ice occurred 60-70% of the time at heights up to 11 km and ice clouds were more prevalent than mixed-phase clouds (Shupe, 2011).



Arctic observatories with permanent ground-based remote sensing measurements include, for example, the French-German
Arctic Research Base AWIPEV in Ny-Ålesund, Svalbard, (78.55°N, 11.56°E) (Hoffmann et al., 2009; Nomokonova et al., 2019; Nakoudi et al., 2021b), the Atmospheric Radiation Measurement (ARM) North Slope of Alaska (NSA) site near Barrow, Alaska, (71.3°N, 156.6°W) (Dong and Mace, 2003; Dong et al., 2010), Summit station, Greenland (72.6°N, 38.5°W) (Shupe et al., 2013; Miller et al., 2015), and Eureka, Canada (80.0°N, 86.42°W) (de Boer et al., 2009).

Cirrus cloud occurence shows strong variations across the Arctic sites and a strong seasonal cycle. Nomokonova et al. (2019)
estimated the occurrence of single-layer ice clouds in Ny-Ålesund to 15-20% in winter and spring, but less than 5% in summer and fall. On the other hand, ice cloud occurence at Eureka varies between 35% in summer and up to 70% in winter (Shupe, 2011).

In addition to the permanent observatories, there have been intensive measurement campaigns with durations of several weeks up to one year including the Surface Heat Budget of the Arctic Ocean project (SHEBA) (Uttal et al., 2002), the Mixed-
Phase Arctic Cloud Experiment (M-PACE) (Verlinde et al., 2007) and the Multidisciplinary drifting Observatory for the Study of Arctic climate (MOSAiC) expedition (Shupe et al., 2022).

Here, we present statistics of cold cloud properties in the Norwegian Arctic as observed by ground-based and spaceborne lidars for the period 2011-2017. The cloud observations were conducted on Andøya (69.3°N, 16.0°E) and focus on the properties of mid- and high-level mixed-phase and cirrus clouds in this region.

The instrumentation and methods with a special focus on the ground-based lidar are described in Section 2. We demonstrate the capabilities of the ground-based lidar in observing cold clouds based on two case studies focused on (1) a cirrus cloud, and (2) a mixed-phase cloud (Section 3). For the cirrus cloud case, we compare the ground-based measurements with collocated observations from the spaceborne lidar onboard CALIPSO for validation. Both platforms are used independently to compute cold cloud statistics for cloud top temperature, cloud top and cloud base heights in Section 4. We discuss the results from both
case studies and the statistics in Section 5 and summarize our conclusions in Section 6.

## 2   Instrumentation and methods

This section is split into a description of the ground-based lidar system in Subsection 2.1, including the methods for processing its raw data, and a short description of the satellite-based instruments and the used data product in Subsection 2.2.

### 2.1   Ground-based lidar

The lidar is part of the Arctic Lidar Observatory for Middle Atmosphere Research (ALOMAR) and co-located with other lidars specialized on the middle and upper atmosphere. It has been in operation since 2005, while the observatory itself was opened in 1994 (Skatteboe, 1996). The tropospheric lidar is part of the European Aerosol Research Lidar Network (EARLINET, Pappalardo et al. (2014)) and is participating in validation activities for satellite missions such as the Atmospheric Dynamics Mission Aeolus (ADM-Aeolus) (Stoffelen et al., 2005).





The monostatic, biaxial system is operating with a pulsed Nd:YAG solid state laser as emitter (primary wavelength: 1064 nm, second and third harmonic: 532 nm, 355 nm; laser repetition frequency 30 Hz) and a Newtonian telescope as receiver. The detection channels include the three emitted wavelengths for elastic scattering and one for Raman scattering at 387 nm. At 532 nm the outgoing light is linearly polarized and the receiver has been equipped with orthogonal and parallel polarization channels since 2011. Also, there are two simultaneous detection channels for every wavelength (except for 387 nm): the analogue mode

for stronger signals, especially in the near range, and the photon counting mode for weaker signals, mostly for the far range. The two channels can be joint through a gluing algorithm, but since we only consider relatively high clouds in this study, we generally use the photon counting signal only. The range resolution of the lidar is 7.5 m, and the time resolution used in this study is 67 s. A more detailed technical description of the instrument can be found in Frioud et al. (2006).

The collected raw data has to undergo several technical corrections before the signal can be physically interpreted. These

include: (1) dead-time correction for the photon counting channels (accounts for the statistical loss of photons in photon-counting mode due to limitations in detection speed), (2) background subtraction (we consider the signal above 40 km as background and subtract the average of this altitude region from the data), and (3) range correction (accounts for the quadratic decrease of the signal with distance). The processed product is the total attenuated backscatter in arbitrary units which is then used in a cloud detection algorithm. In case studies, we additionally use lidar constants computed by the EARLINET Single

Calculus Chain (D'Amico et al., 2015) to convert total attenuated backscatter from arbitrary units into $(m·sr)^{-1}$.

Besides analyzing macroscopic cloud properties statistically, we use the linear volume depolarization ratio to identify regions of different cloud thermodynamic phase and particle composition inside the cloud during the case studies. The linear volume depolarization ratio $\delta$ is defined as the ratio of cross-parallel polarized backscatter $\beta_\perp$ to parallel polarized backscatter $\beta_\parallel$.

$$\delta = \beta_\perp / \beta_\parallel \tag{1}$$

For calibrating the polarization-filtered signals against each other, we use the $\pm 45^o$-method described in Freudenthaler et al. (2009).

Another important property for the cloud's radiative properties is the cloud optical depth $\tau$. The possibility to calculate cloud optical depth from lidar data is limited to optically thin clouds and we follow a technique originally developed for micropulse lidars of the ARM program at the U.S. Department of Energy (Lo et al., 2006; Comstock and Sassen, 2001):

$$\beta_c(z) = \frac{G(z_0, z)}{1 - \frac{2\eta}{k} \int_{z_0}^{z} G(z_0, z')dz'} - \beta_m(z) \tag{2}$$

with    $$G(z_0, z) = \beta_m(z_0) \frac{S(z)z^2}{S(z_0)z_0} \cdot \exp\left[2\left(\frac{8\pi}{3} - \frac{\eta}{k}\right) \int_{z_0}^{z} \beta_m(z'')dz''\right] \tag{3}$$

Here, $\beta_c(z)$ and $\beta_m(z)$ are the cloud and molecular backscatter coefficient, respectively, as a function of the altitude $z$. $\eta$ is the multiple scattering coefficient, $z_0$ is a boundary height below the cloud where the air is assumed to be cloud-free, $k$ is the backscatter-to-extinction ratio, and $S$ is the sum of the processed parallel and cross parallel signal. There have been various

approaches to determine the multiple scattering coefficient for cirrus clouds and commonly used values vary between 0.4 (Platt (1973) proposed $0.41 \pm 0.15$ from observations) and 0.9 (upper maximum used by Comstock and Sassen (2001)). For this




study, we decided to use $\eta = 0.8$ in agreement with Lo et al. (2006) and Comstock and Sassen (2001). The backscatter-to-extinction ratio $k$ is varied between 0.01 and 0.2 such that the total backscatter above the cloud is closest to the molecular backscatter. Afterwards, the optical depth $\tau$ of a cloud with cloud base $z_b$ and cloud top $z_t$ is calculated as

$$\tau = \frac{1}{k} \int\limits_{z_b}^{z_t} \beta_c(z) dz \tag{4}$$

The cloud detection is based on an algorithm developed by Gong et al. (2011). It uses only one wavelength and due to its lowest Rayleigh scattering efficiency and therefore highest penetration into a cloud, we apply it to the 1064 nm channel. After smoothing and noise level calculation, the signal is first simplified using a Douglas-Peucker algorithm (Douglas and Peucker, 1973) that identifies points with large gradient changes (vertices). A cloud base is detected if the gradient between two vertices exceeds a certain threshold and the signal is above noise level. The corresponding cloud top is identified as the first vertex with a lower signal strength than the base vertex. The threshold is empirically determined and set to $10^5$. To avoid false identifications, all clouds detected by the algorithm are manually verified.

The cloud top temperatures (CTTs) for the clouds identified by the ground-based lidar are retrieved from nearby released radiosondes (Norwegian Meteorological Institute, 2021) and ERA5 reanalysis data (Hersbach et al., 2020). There have been two radiosonde releases daily in Andenes starting in October 2014, which is only 5 km away from the ground-based lidar. Before, the closest releases were from Bodø (67.28°N, 14.45°E) which is too far away to be considered relevant for routinely retrieving CTTs. Therefore, we use ERA5 reanalysis data for the period before 2014 (downloaded from Hersbach et al. (2018)) and radiosonde data afterwards. In order to compare both methods, we tested their correlation for the years 2015-2017 (see Fig. 1). For ERA5 temperatures, we interpolate the cloud top temperature linearly between the two closest pressure levels. With a correlation coefficient of 0.95, the agreement is generally good, but the interpolated temperatures from reanalysis data show a tendency to be higher than the ones measured by the radiosonde, on average by 1 K. These differences can be attributed to the horizontal displacement of the radiosondes and uncertainties in the ERA5 reanalysis (30 km horizontal resolution, 137 pressure levels between the surface and 80 km altitude).

The available measurement record of observations including depolarization sensitive channels spans from 2011 until today with a maintenance break from April 2013 to July 2015. The lidar can be operated whenever there is no precipitation and the 10-min average wind speed does not exceed 12 m/s. The majority of measurements is made during daytime. Possible implications and biases of measurement routines will be discussed in Section 4.

## 2.2 Spaceborne lidar and radar

We are using data from the cloud profiling radar (CPR) on CloudSat (Stephens et al., 2002) and the Cloud-Aerosol Lidar with Orthogonal Polarization (CALIOP) on the Cloud-Aerosol Lidar and Infrared Pathfinder Satellite Observation (CALIPSO) satellite (Winker et al., 2009). For direct comparison with the ground-based lidar, we use CALIOP level 1 (1B Profile, NASA/LARC/SD/ASDC (2016)) and level 2 (5 km Cloud Layer, NASA/LARC/SD/ASDC (2018)) data products for cloud properties such as backscatter, altitude, and optical depth. CALIOP operates at the same wavelengths of 1064 nm and 532 nm





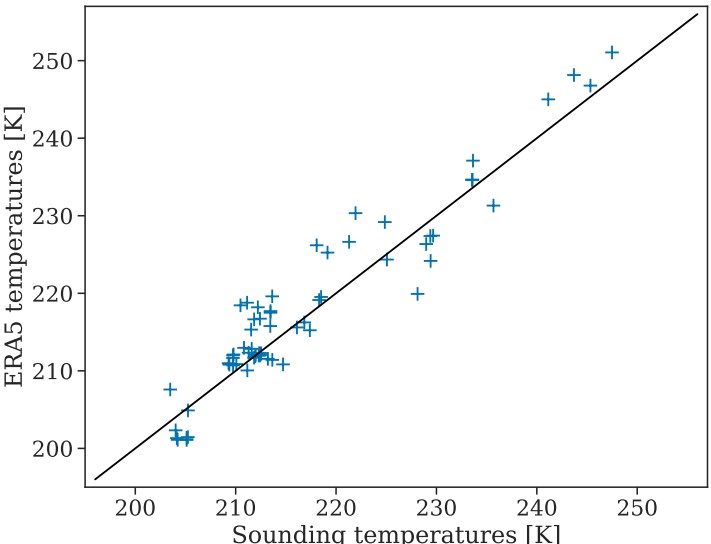

**Figure 1.** Cloud top temperature extracted from ERA5 scattered against cloud top temperature from closest available radiosonde release (Andenes) for all detected cirrus clouds between 2015 and 2017. The black 1:1-line indicates exact agreement and makes a slight bias of ERA5 data visible towards warmer temperatures. The average difference is 1 K.

as the lidar at ALOMAR. Here, the vertical resolution in the relevant altitude region above 8 km is 60 m, the horizontal resolu-

tion is 1 km. For the phase discrimination between cirrus, mixed-phase and liquid clouds, we use the 2B-CLDCLASS-LIDAR data product (Sassen et al., 2008), which utilizes the different sensitivities of the radar and lidar to liquid droplets and ice crystals. Beside the cloud phase, we use cloud top and base height information and, for each cloudy profile, we retrieve the cloud top temperature from the ECMWF-AUX dataset (version $P_R05$) that uses ancillary European Centre for Medium-Range Weather Forecast (ECMWF) state variable data interpolated to each CPR vertical bin. Apart from gaps in the satellite dataset

(January 2011, January-April 2012, and June, July, and September 2017) we analyze the period 2011-2017.

### 2.3 Comparison of ground-based and spaceborne observations

For the statistical analysis, we split the available data from the ground-based lidar into 30 minute measurement intervals (shorter measurements count as one interval as well), which results in a total number of 1366 measurements between 2011 and 2017. We include all satellite overpasses within a 2°x2° box around ALOMAR in the analysis, which corresponds to an extent of

approximately 80 km in zonal and 220 km in meridional direction. This increases the number of satellite overpasses, resulting in 48873 individual profiles being used for the statistical analysis of cold cloud properties.

We limit this analysis to single-layer clouds, to avoid attenuation by the lidar when penetrating multiple cloud layers, which would induce a bias in the statistics due to the opposite up- and downward viewing configurations of the two lidar systems. A





cloud scene observed from the ground-based lidar is considered multi-layered if there is a cloud-free region of at least 200 m
vertical distance between two cloudy layers. Otherwise, the scene is regarded as a single-layered cloud.

## 3   Case study results

To demonstrate the capabilities of the ground-based lidar in observing cold clouds, we present two case studies focusing on (1) a
cirrus cloud, and (2) a mixed-phase cloud. The cirrus cloud case provides the opportunity to directly compare the ground-based
lidar with measurements from the spaceborne lidar CALIOP. For the mixed-phase cloud case, we use the lidar to distinguish
between liquid droplets and ice crystals, giving insights into the vertical phase distribution of the cloud.

### 3.1   Cirrus cloud case

On 1 April 2011, CALIPSO passed over the ALOMAR lidar at 11:11 UTC (13:11 LT), while both lidars were measuring a
cloud layer between 9.6 and 11.5 km altitude. The satellite ground track is shown in Fig. 2d, crossing the island of Andøya
with a horizontal distance to ALOMAR of 2 km north-east of the mountain Ramnan.
180       On that day, the Norwegian west coast was located between a low pressure system west of Iceland and a high pressure
system centered over Svalbard. The cirrus clouds observed here were located in a region between a warm front in the north,
that had passed Andøya the day before, and a vanishing occluded front, that reached from the Atlantic east of Greenland to
Southern Norway (Met Office, 2021).
       The ground-based lidar was running from 09:53 to 11:20 UTC, followed by a depolarization calibration measurement. From
satellite, we use data from the time when the ground track is located in a geographical box of 2°x2° in meridional and zonal
direction around ALOMAR. This is the case from 11:11:09 until 11:11:37 UTC, i.e. for a total duration of 28 seconds.
       The total attenuated backscatter from ground- and spaceborne lidar is shown in Fig. 2a and 2c. The total attenuated backscatter from the ground-based lidar ranges from 1 to $2.5\cdot10^{-6}\,(\mathrm{m\cdot sr})^{-1}$ in the cloud around the time of the CALIPSO overpass
(black vertical line). In terms of vertical structure, the lowest backscatter values inside the cloud are found around 11 km,
the layer intensifies from there towards both top and base. From the spaceborne lidar, the total attenuated backscatter ranges
between 1 - $4\cdot10^{-6}\,(\mathrm{m\cdot sr})^{-1}$ and the vertical substructure is less clear, but still recognizable especially at the latitude closest
to ALOMAR until around 69.6°N. Here, the retrieved cirrus cloud base and top heights are 9.6 km and 11.4 km, respectively.
This is in good agreement with the ground-based lidar. Taking into account temperature data from the closest available radiosonde release (from Bodø, 67.28°N, 14.45°E, 11:10 UTC), we see that the temperature at this altitude was -60 °C and
lower, i.e. well below the limit for homogeneous freezing (e.g., Heymsfield and Sabin, 1989). From the vertical temperature
profiles we estimate the tropopause to be located at about 11.0 km (from radiosonde) or 10.6 km (from reanalysis data) at a
temperature of -70 °C. Thus, the cirrus cloud is extending well into the tropopause, dehumidifying the upper troposphere and
lower stratosphere region through ice crystal growth and sedimentation (e.g., Kärcher, 2005).
       The linear volume depolarization ratio from the ground-based lidar is shown in Fig. 2b and ranges from 0.2 to 0.3, indicating
thin plate-like particles (shape ratio length/diameter < 0.1) and intermediate and irregular shapes with shape ratios up to 0.5



**Figure 2.** Cirrus cloud measurement on 1 April 2001: Total attenuated backscatter at 532 nm from the ALOMAR lidar (a) and CALIOP (c), and linear volume depolarization ratio from the ground-based lidar (b). The closest available radiosounding from Bodø is shown in (d), together with a map showing the groundtrack of the satellite (blue line) and the position of the groundbased lidar (red cross). The black lines show the overpass time in (a), (b) and the closest location during the overpass in (c). The vertical and temporal resolution of the ALOMAR lidar are 7.5 m and 67 s, respectively. The satellite resolution is 60 m in vertical and 1 km in horizontal direction. In addition, the satellite backscatter is smoothed by a Gaussian filter.





(categories I and II in Noel et al. (2002)). From the satellite products, layer integrated depolarization ratio is available and has values between 0 and 0.4 over the displayed period (not shown) which covers the range observed by ground-based lidar. A more detailed comparison is not possible as the noise level of the linear volume depolarization ratio from satellite is too high.

To compare the cloud optical depths $\tau$ as retrieved from both platforms, we estimated the backscatter-to-extinction ratio with
$k = 0.2$, which yields the best agreement with molecular backscatter above the cloud. Averaging over the time interval 09:55 to 11:20, the ground-based lidar gives an optical depth $\tau$ of 0.07±0.02. At the location where the satellite groundtrack has the shortest distance to ALOMAR, the spaceborne lidar retrieves the same value of $\tau = 0.07 \pm 0.02$. Thus, the retrieved cloud optical depths show a very good agreement. The observed cloud is an optically thin cirrus cloud according to the classification by Sassen and Cho (1992) (0.03<$\tau$<0.3). This is also the most common category found at Ny-Ålesund, Svalbard, with 73%
occurrence (Nakoudi et al., 2021a), and at the subarctic site of Kuopio, Finland, (62.74°N, 27.54°E) with 71% occurrence (Voudouri et al., 2020).

### 3.2 Mixed-phase cloud case

The second case that is selected for detailed analysis is an altocumulus mixed-phase cloud as shown in the image in Fig. 3a. It was observed on 24 August 2017 from 10:10 to 10:40 UTC and located at 5.2 to 5.4 km altitude. The general weather situation
in Northern Norway that day was influenced by two high pressure systems, one located over Greenland, the other over the Barents Sea extending over the Atlantic towards Iceland. At the same time two low pressure systems were located northwest of the British islands and close to St. Petersburg, Russia (Met Office, 2021). The synoptic situation results in fields of scattered clouds along the Norwegian coast, mostly of orographic origin over land (see Fig. 3b).

The radio sounding from Andenes at 11:04 UTC (Fig. 3c) reveals a temperature of -24 to -26 °C in the relevant altitude
region. It increases 2 K right above 5.6 km, at the same altitude where the dew point temperature drops more than 10 K. This indicates a sudden decrease in humidity marking the border between two air masses of which the lower one is connected to the cloud and the upper one is warmer and dry above the cloud top. Further interpretation of the sounding data is not possible as the radiosonde did not penetrate the cloud, but the air mass behind it. The pronounced cloud boundary can be seen in Fig. 3a.

The total attenuated backscatter and the linear volume depolarization ratio from the ground-based lidar are shown in Fig. 4.
From the backscatter signal, the distinct cloud boundaries at 5.2 and 5.4 km altitude with falling hydrometeors below become apparent. Differences in linear volume depolarization ratio imply different shapes of the cloud particles (Noel et al., 2002), at least as long as single scattering is concerned. Hexagonal ice crystals typically lead to linear depolarization ratios from 0.2 to 0.5, depending on the aspect ratio, while single-scattering from spherical water droplets does not induce any polarization change. The depolarization ratio in the mixed-phase cloud case shows a clear separation in three regions (see Fig. 4b): the
center of the cloud with values below 0.1 can be clearly separated from the cloud top and a large area below cloud that extends down to 4.5 km (both with values up to 0.4). Thus, spherical liquid water droplets dominate in the region around 5.2 km with $\delta < 0.1$, whereas the high depolarization values between 4.5 and 5.2 km altitude can be attributed to falling ice crystals (virga). The increasing depolarization ratio inside the cloud body towards the cloud top can be attributed to multiple scattering by liquid water droplets, since the cloud is optically thick such that there is no signal coming back from above the cloud. Hu et al. (2006)





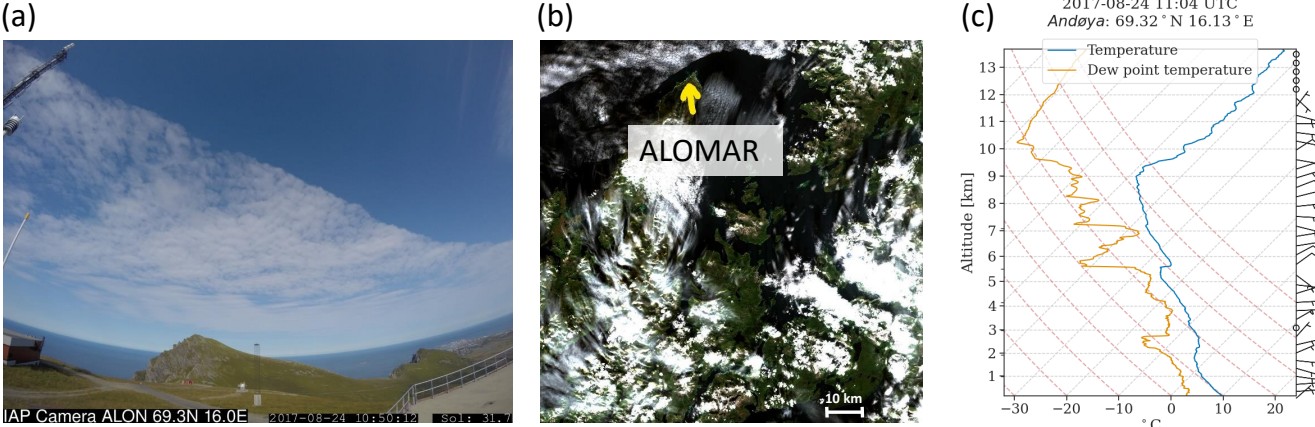

**Figure 3.** Images of the cloud field probed by lidar at 10:50 UTC as seen (a) from inside the ALOMAR observatory and (b) from the Sentinel-2 satellite (Drusch et al., 2012) (true color image). In the satellite image, the northern tip of the island is marked by the yellow arrow, the cloud field probed by the lidar is right north of the island. The border between cloud and clear sky visible in both pictures moves through the lidar's field of view around 10:40 UTC. Temperature, dew point and wind profiles from a radiosonde released in Andenes at 11:04 UTC on 24 August 2017 are given in (c).

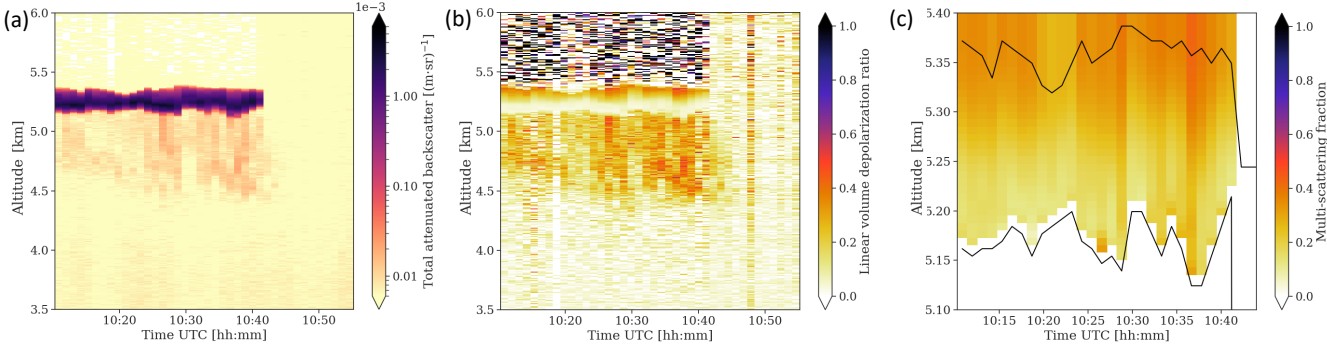

**Figure 4.** (a) Total attenuated backscatter (TAB), (b) linear volume depolarization ratio at 532 nm, and (c) accumulated multiple scattering fraction inside the liquid cloud body. The black lines indicate cloud base and top of the liquid cloud body as identified from the combined parallel and cross-parallel signal. The multiple scattering fraction is accumulated from cloud base.

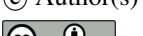



presented a relationship between accumulated multiple-scattering fraction in water clouds and accumulated depolarization ratio $\delta_{acc}$. We retrieve the altitude of the (assumed) liquid cloud base and top from the gradient in the attenuated backscatter signal, and take this cloud base as a starting point for the accumulated depolarization and multiple-scattering ratios. Applying the formula $A_S = 0.999 - 3.906\delta_{acc} + 6.263\delta_{acc}^2 - 3.554\delta_{acc}^3$ from Hu et al. (2006) results in the profile of the multiple scattering fraction within the cloud as shown in Fig. 4c. The fraction of multiple scattering increases from around 15% at cloud base

to up to 40% at cloud top. Note that this calculation assumes that the depolarization signal is entirely explained by multiple scattering from spherical water droplets. Hence, the multiple scattering fraction of 15% already at cloud base indicates an additional influence by ice crystals within the predominantly liquid cloud layer. This is typical for a liquid-topped mixed-phase cloud, where small ice crystals are formed at the cloud top and then fall through the liquid part of the cloud. The observed structure is in accordance with, for example, in-situ observations from aircraft by Barrett et al. (2020) and ground-based lidar

observations by Engelmann et al. (2021). They found ice production within the supercooled layer at temperatures of -30°C and -28.5°C, respectively, and ice virgae below. Thus, we conclude that there are active ice nucleating particles at temperatures around -25°C, but in insufficient amounts for complete glaciation of the cloud.

## 4   Cold cloud statistics

The statistical analysis of cold cloud properties uses all data from the ground-based lidar at ALOMAR since the installation of

the depolarization channel in 2011 until 2017 as well as spaceborne lidar data spanning the same period.

    We define cirrus clouds as all single-layer clouds with cloud base heights between 4000 and 12000 m and a cloud top temperature below -20 °C (253.15 K). This is based on Sassen et al. (2008) and Heymsfield et al. (2017), which showed from satellite observations that cirrus clouds in the Arctic are mostly limited to an altitude range between 4-12 km.

    To test this definition, we applied it to all 25779 single-layer clouds detected from CloudSat/CALIPSO during the study

period. Of these, 3838 clouds were identified as cirrus clouds. Figure 5c shows the location of these cirrus cloud profiles within the 2° x 2° box around ALOMAR. Using the additional phase information, we find that 95% of these clouds were indeed pure ice clouds (3638 cases, see Fig. 5b). The remaining 5% consisted of mixed-phase clouds (187 cases) and pure liquid clouds (13 cases). This confirms that the cirrus cloud definition captures mostly pure ice clouds, however it cannot be ruled out that some of the cirrus cloud cases identified from the ground-based lidar might include some mixed-phase clouds. A further phase

discrimination from the ground-based lidar for this possible mixed-phase cloud influence is beyond the scope of this study. Due to this slight ambiguity, we hereafter refer to these predominantly ice clouds as cold clouds. Furthermore, we conclude that INP concentrations generally are high enough to glaciate single-layer clouds at temperatures below -20°C.

    As can be seen in Fig. 5a, the monthly occurrence of single-layer cold clouds varied between 4% and 13% showing no clear seasonal dependence. On average, 8% of all satellite profiles were identified as single-layer cold clouds. With a total

occurrence of 51% for single-layer clouds of all heights and phase compositions, this corresponds to 15.4% of all single-layer clouds being cold clouds in our definition. This is less than one out of four as reported by Nomokonova et al. (2019) for Ny-Ålesund, Svalbard (36% total occurrence of single-layered clouds, thereof 9% pure ice clouds). The number of cirrus cloud





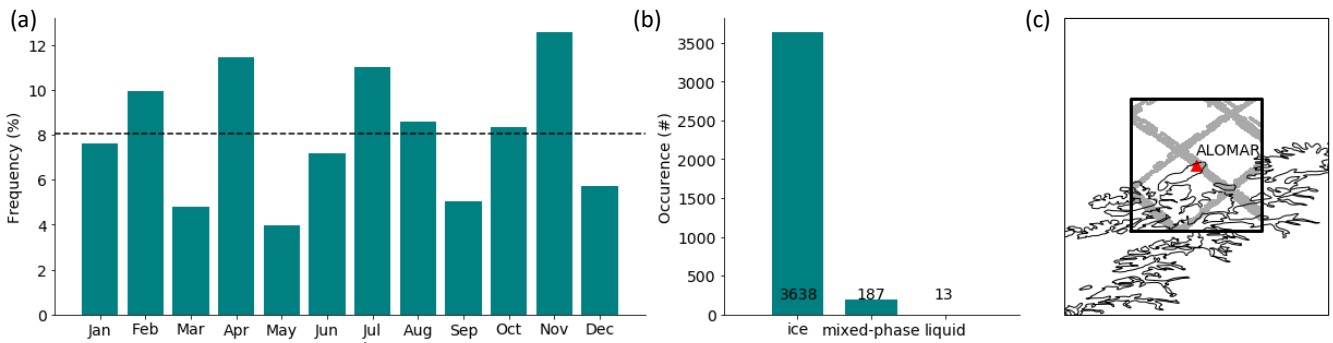

**Figure 5.** (a) Monthly single-layer cold cloud occurrence from satellite. (b) Distribution of cloud phase, given in number of observed single-layer cold clouds. (c) Spatial box from which cloud measurements are taken for the analysis: The ground-based lidar at ALOMAR is indicated by the red triangle, the box extends two degrees around ALOMAR (one degree in each direction). Satellite overpasses with cloud detection are indicated by grey dots.

observations from the ground-based lidar also shows no seasonal trend. However, the ground-based record is not continuous due to operation hours and weather limitations: measurements are not possible in case of precipitation or average wind speeds

exceeding 13 m/s due to local instrument safety restrictions. Thus, the cold cloud occurrence as seen from the ground-based lidar has a bias towards higher values and is not shown here as it cannot be compared to the spaceborne lidar.

Nevertheless, the macroscopic cold cloud properties (cloud top and base height) for both the ground-based and satellite observations can be compared and are displayed in Figure 6. The corresponding seasonal averages can be found in Tab. 1. The ground-based lidar records cloud top heights between 5045 m and 13130 m (mean: 9240 m) and cloud base heights between

4040 m and 11090 m (mean: 6970 m) with a pronounced annual cycle. There are distinct increases in height from January to February, May to June and August to September as well as decreases from February to March and September to October. In general, there is a trend towards higher cold clouds in summer and fall compared to winter and spring. The largest monthly averaged altitudes (both cloud base and top) are recorded in September, the lowest in January. The general trend of higher cold clouds in fall than in winter and spring is also apparent in the results from the spaceborne observations (see Fig. 6b).

However, the monthly variability is less pronounced than for the ground-based measurements, indicating an influence of the irregular observation times for the ground-based lidar. Cloud top heights as retrieved from the satellite are also slightly lower (4150 m-12490 m, mean: 8915 m) than for the ground-based lidar. The cloud base height from satellite varies between 4030 m and 11890 m (mean: 6895 m), which is more similar to the ground-based observations. However, this is expected due to the cloud type detection algorithm being dependent on the cloud base height. The large standard deviations for cloud top and base

heights as visible in Fig. 6 indicate a larger case-to-case than monthly variability for both platforms. Moreover, the average vertical extents of the cold clouds are similar with 2270 m (ground-based) and 2020 m (spaceborne), respectively.



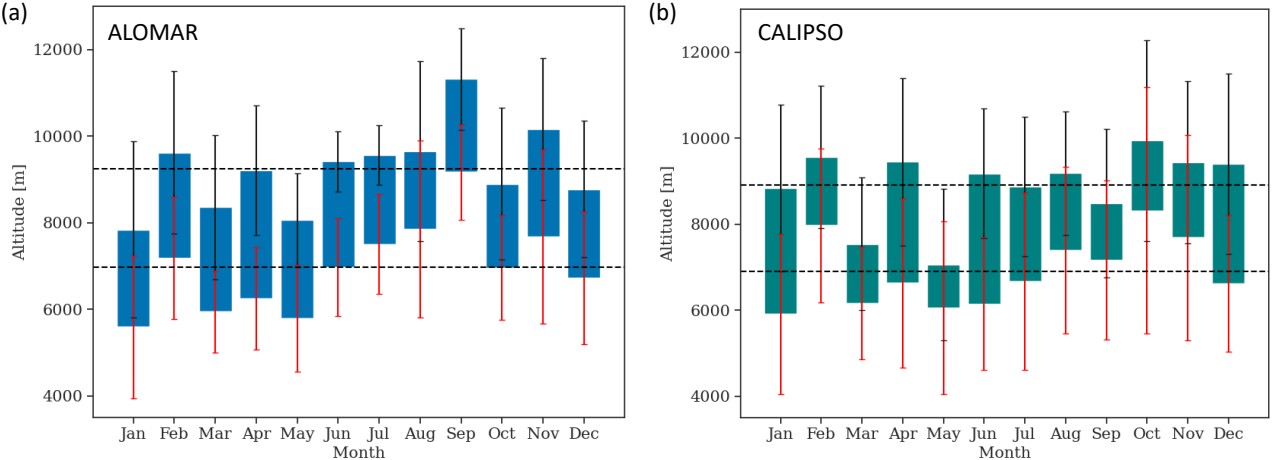

**Figure 6.** Monthly mean thickness of cirrus clouds observed from (a) the ground-based lidar at ALOMAR and (b) CALIOP. The error bars indicate the standard deviation of the computed top and base heights. The dashed black lines show all-year average.

We show histograms as well as monthly averages of the cloud top temperatures from both lidars in Fig. 7. The distributions of observed cloud top temperatures are similar from both platforms: the registered cold clouds showed CTTs between 201-253 K (ground-based) and 196-252 K (spaceborne) with a pronounced maximum of cold cloud occurence at around 220 K

(ground-based) and 212 K (satellite). The distribution from ground-based lidar has a second (lower) maximum at 211 K as well, closer to the maximum observed from satellite. The main difference between both distributions is the total number of measurements, which leads to the more patchy histogram for the ground-based observations, especially towards the edges of the distribution. Furthermore, the observed cold cloud CTTs show a similar annual cycle from both platforms (see Fig. 7c and d). The highest CTTs were registered in summer (between May and August) with up to 230 K, whereas CTTs are lowest in

the winter months (214 K in December for ground-based, 208 K in February for spaceborne). For the satellite measurements, this low CTT in February coincides with the high mean CTH in February (see Fig. 6b). However, even though the cold clouds in December show the coldest CTTs from the ground-based lidar, their corresponding CTH is not the highest throughout the year. This can partly be explained with the lower temperatures throughout the troposphere in the winter. However, similarly low CTTs in September correspond well to the highest CTHs registered in that month (see Fig. 6a).

**5  Discussion**

First, we put our results in relation with long-term ground-based observations of clouds in the Arctic, before comparing with spaceborne instrument studies. The all-year averaged total cloud occurrence at 4 km altitude varies between around 15% (Barrow, Alaska) and up to 30% (SHEBA, ship-based observatory in the western Arctic Ocean) and decreases with altitude to less than 1% above 10 km altitude (Shupe, 2011). These values are higher than our finding of 8% cold cloud occurrence, and



| Variable, instrument | winter (DJF) | spring (MAM) | summer (JJA) | autumn (SON) | all-year |
|---|---|---|---|---|---|
| **Occurrence** | | | | | |
| satellite [#] | 626 | 1064 | 1250 | 898 | 3838 |
| satellite [%] | 7.8 | 6.9 | 8.8 | 8.0 | 8.0 |
| **Cloud top temperature [K]** | | | | | |
| satellite (ECMWF) | 211.8 | 221.3 | 227.3 | 220.4 | 220.2 |
| ground-based | 217.1 | 224.1 | 227.0 | 218.3 | 221.6 |
| **Cloud base height [m]** | | | | | |
| satellite | 6832 | 6286 | 6740 | 7724 | 6896 |
| ground-based | 6501 | 5995 | 7427 | 7938 | 6965 |
| sat./ground-based average | | | | | 6930±35 |
| **Cloud top height [m]** | | | | | |
| satellite | 9267 | 8017 | 9082 | 9295 | 8915 |
| ground-based | 8746 | 8545 | 9527 | 10125 | 9236 |
| sat./ground-based average | | | | | 9075±160 |

**Table 1.** Seasonal and all-year average of occurrence, temperature and height of clouds for the two datasets from the ground- and satellite-based lidar. The occurrence for satellite is the total number of detections between 2011 and 2017 during the respective season.

the difference can be explained by the restriction to single-layer clouds, while Shupe (2011) takes into account multi-layered clouds as well. On the other hand, our result is still larger than the annual mean cirrus occurrence of $2.7 \pm 1.8\%$ reported from Ny-Ålesund by Nakoudi et al. (2021a), which is meaningful as their value is according to the authors negatively biased due to the very strict criteria for reliable detection. In terms of geometrical properties, Nakoudi et al. (2021a) also find a mean thickness of 2 km and higher cloud bases during summer and autumn than during winter and spring. Relatively large variations in

geometrical thickness with a tendency towards thicker layers in winter seem to be common at Northern high latitudes (Nakoudi et al., 2021a; Devasthale et al., 2011).

In the following, we compare our results for cold cloud occurrence with earlier estimates of cirrus cloud frequency at high latitudes from CALIPSO and CloudSat as done by Mace et al. (2009), Nazaryan et al. (2008) and Gasparini et al. (2018). The analysis by Mace et al. (2009) and Nazaryan et al. (2008) are restricted to the first year of observations from CALIPSO

and CloudSat. While Mace et al. (2009) address hydrometeor layers of all altitudes and compositions, Nazaryan et al. (2008) focus on cirrus clouds. They find values of cirrus cloud occurrence between below 20% and nearly 30%, depending on the season and how multi-layered clouds are treated in the analysis. These occurrences are more than double of the satellite values presented in our study. However, it is important to note that all of these studies include observations with multiple layers. For their "single-layer" statistics, only the top layer is considered (Nazaryan et al., 2008), while in our study only single-layer

observations are considered.





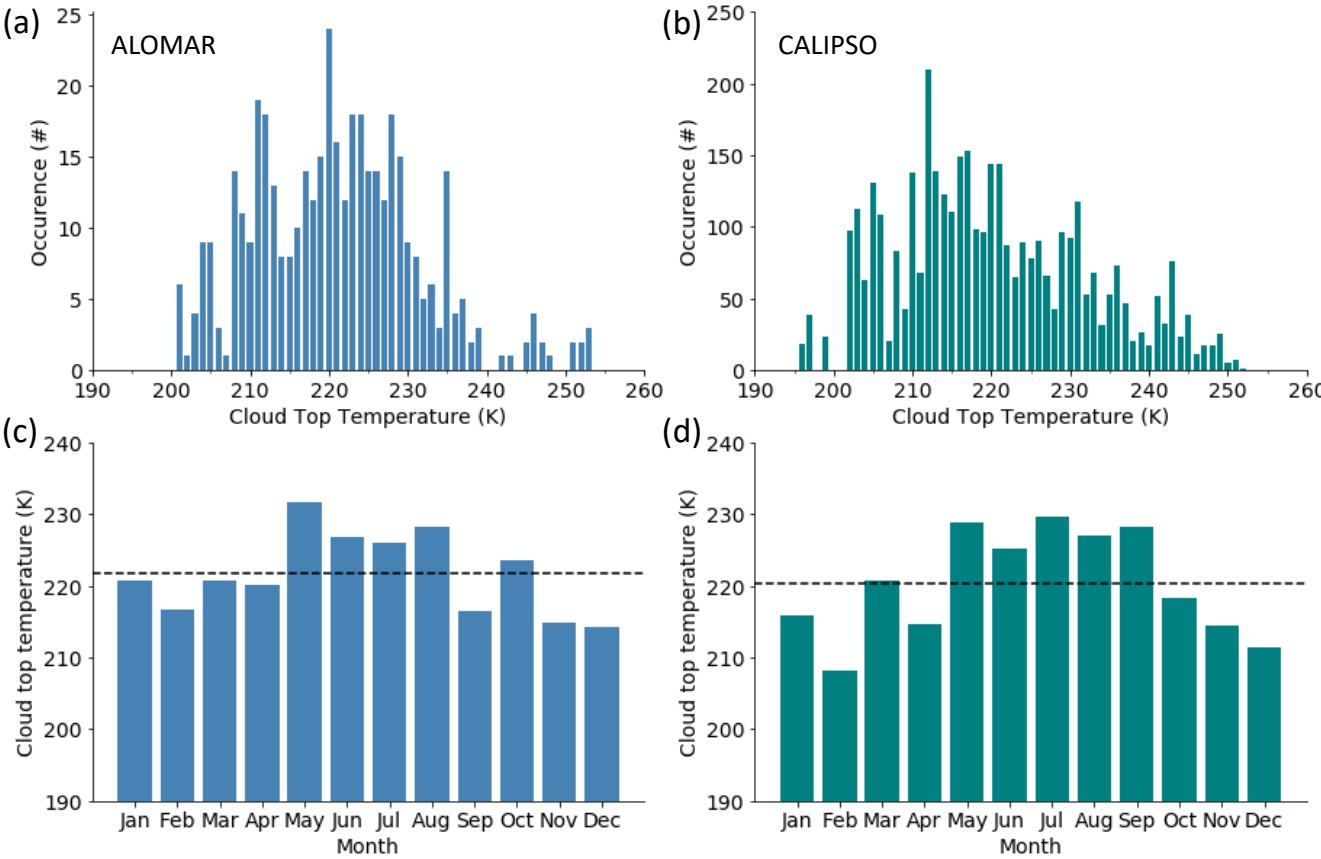

**Figure 7.** Upper row: Histogram of cloud top temperatures for all cirrus detections from (a) the ground-based lidar and (b) the spaceborne lidar between 2011 and 2017. For the ground-based lidar, the temperatures are interpolated from reanalysis data (ERA5) on pressure levels until 2014 and taken from radiosondes thereafter. Lower row: Monthly averaged cloud top temperature from (c) ground-based and (d) spaceborne lidar.

## 6 Conclusions

We use the record of the tropospheric lidar at the ALOMAR observatory on Andøya in the Norwegian Arctic to retrieve macroscopic (cloud top and base height) and microphysical cloud properties. In analysing (1) a cirrus cloud and (2) a mixed-phase cloud case, we demonstrate the capabilities of the ground-based lidar in observing cold cloud properties. Collocated observations from the spaceborne lidar onboard CALIPSO allow for a direct comparison of both lidars for the cirrus cloud case. We then compare the statistics of cold cloud properties in the Norwegian Arctic as observed from the ground-based and spaceborne instruments between 2011-2017. For this, we define cold clouds as all single-layer clouds with cloud base heights between 4000 and 12000 m and a cloud top temperature below -20 °C. Applying this definition to the satellite profiles, we





find that 95% of these clouds were pure ice clouds. Therefore, ice nucleating particle concentrations are often sufficient to completely glaciate single-layer clouds at the given temperatures.

Our main conclusions are:

1. Observations of an optically thin cirrus cloud agree well between ground-based and spaceborne lidar in terms of the cloud height and optical depth. Cloud height deviations are on the order of 100 m or less and the difference in retrieved optical depth is below 10%.

2. Polarization sensitive measurements in combination with multiple scattering considerations from the ground-based lidar can be used to determine cloud structure and vertical phase composition as demonstrated for a mixed-phase altocumulus cloud.

3. Between 2011 and 2017, on average 8% of all satellite profiles were identified as single-layer cold clouds (corresponding to 15.4% of all single-layer clouds). Their average thickness is 2.0 km. No clear seasonal cycle for the cold cloud occurrence could be identified from the satellite measurements.

4. The ground-based lidar records mean cold cloud top and base heights of 9.2 km and 7.0 km, respectively, with a trend towards higher clouds in summer and fall compared to winter and spring. The mean cold cloud top and base heights as retrieved from the spaceborne lidar are 8.9 km and 6.9 km and, thus, slightly lower than for the ground-based lidar. The seasonal variability in cloud thickness and height is smaller than the case-to-case variability.

5. Cold clouds in the Norwegian Arctic are on average between 1 and 2 km higher in autumn than in spring, while winter and summer show intermediate values. This is confirmed by both ground-based and spaceborne observations.

6. For both platforms, the retrieved cloud top temperatures show similar distributions and a good agreement in their annual cycle with warmer CTTs in summer.

7. Cold cloud properties in the Norwegian Arctic agree well with observations from other Arctic sites. Geometrical properties are very similar to Ny-Ålesund, Svalbard, and occurrence is within the range found at sites in Alaska, Canada, the Arctic Ocean, and Svalbard.

Limitations in the applicability of the lidar for mixed-phase cloud research are mainly connected to the restriction to elastic scattering channels during daylight measurements. When using a single field-of-view lidar and elastic channels only, a more detailed study of the microphysical processes requires complementary observational data from radiosondes and sensitivity studies with radiative transfer simulations in order to account for multiple-scattering effects.

The ground-based lidar at ALOMAR is still in operation and its long-term installation provides an opportunity to study changes in cold cloud properties in the rapidly changing Arctic.



*Data availability.* The used satellite data is available from the CloudSat Data Processing Center https://www.cloudsat.cira.colostate.edu/ (CloudSat) and the NASA Langley Research Center - Atmospheric Sciences Data Center https://asdc.larc.nasa.gov/ (CALIOP). ERA5 tem-
perature data is available from the Copernicus Climate Data Store https://cds.climate.copernicus.eu/ (Hersbach et al., 2018) and radiosounding profiles are available from the Norwegian Meteorological Institute https://thredds.met.no/thredds/catalog.html (Norwegian Meteorological Institute, 2021). Ground-based lidar data can be made available upon request.

*Author contributions.* BS, TS and TC designed the study. IH and MG provided the ground-based lidar raw data. IH provided the framework for the analysis of the raw data. BS performed the data analysis and interpretation for the case studies, as well as statistics from the ground-
based lidar. MG contributed to the interpretation of the results. TC performed the analysis of statistics from the spaceborne lidar. BS and TC prepared the manuscript with help of TS.

*Competing interests.* The authors declare that they have no conflict of interests.

*Acknowledgements.* We gratefully acknowledge the funding by the European Research Council (ERC) through Grant StG758005. We thank all lidar operators who contributed to the ground-based dataset. In addition, we gratefully acknowledge the donation of lasers for the tro-
pospheric lidar system at ALOMAR by the Leibniz-Institute of Atmospheric Physics at the University of Rostock and thank Jens Fiedler and Götz von Cossart for their technical support. From the same institute, we thank Gerd Baumgarten for providing the cloud image in Fig. 3a. Furthermore, we thank Malin Abrahamsen (Andøya Space) for her work on the code for cloud detection. ERA5 data was downloaded from the Copernicus Climate Change Service (C3S) Climate Data Store. CALIOP level 1 and 2 data products were obtained from the NASA Langley Research Center - Atmospheric Sciences Data Center. The standard CloudSat and CALIPSO data products (version R05) used in
this study (2B-CLDCLASS-LIDAR, ECMWF-AUX) were downloaded from the CloudSat Data Processing Center's (at Cooperative Institute for Research in the Atmosphere, Colorado State University, Fort Collins) website (http://www.cloudsat.cira.colostate.edu/).





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
