# Peer review of "Observations of cold cloud properties in the Norwegian Arctic using ground-based and spaceborne lidar"

_Atmospheric Chemistry and Physics, 2021_

## Author Comment (AC1)

RC 1 answer:

This manuscript compares cloud macrophysical properties from ground-based and spaceborne lidar observations. For two case studies and a 7-year long data set, the cloud base and top heights, and the phase of the clouds observed over Andenes in the Norwegian Arctic were derived. By means of thermodynamic profiles, the temperature at cloud top was estimated. For the ground-based lidar, the closest radiosonde or ERA-5 was applied, while for the spaceborne system the ECMWF-AUX data product was used. Additionally, the phase separation capability of a polarization lidar is highlighted. The manuscript is well structured and it provides a valuable contribution to the study of Arctic clouds. The paper is of interest to the community, especially the comparison between the ground-based and spaceborne lidar systems is of importance and it should be published after some major revisions were made.

> We want to thank Reviewer 1 for the insightful comments and remarks that helped improving our manuscript considerably. Please find the point-by-point responses to your comments below. When citing the text from the revised manuscript in the responses, we refer to page and line numbers in the new manuscript. If only single words are changed in sentences, they are given in italic font.

The reason to apply different methods for the estimation of the cloud top temperature is not clear to me. If the goal was to assess the ECMWF-AUX data product, this should be made more clear in the manuscript. However, if the objective was to compare the observations of the two lidar systems, the same approach to derive the cloud top temperature should be applied. In this case, I do not see a reason, to not also not apply the radiosonde/ERA-5 approach to derive the cloud top temperature for the spaceborne observations.

> Thank you for the comment. It is true that the use of different reanalysis products has an influence on the cloud top temperature comparison between the spaceborne and ground-based observations. We do not aim at an assessment of the quality of the ECMWF-AUX product. Nevertheless, ECMWF-AUX has been specifically designed to provide profiles of temperature from atmospheric reanalysis interpolated on the time and location of the CloudSat/CALIPSO overpass, which makes it the first choice to use in combination with the phase retrieval from the 2B-CLDCLASS-LIDAR product. However, we agree that using different reanalysis products and displaying the cloud top temperature distributions with 1K bins in Figure 7 is problematic (see also your comment below). Thus, we decided to increase the bin size to 2.5 K, thus enabling us to draw conclusions about the general CTT distribution of cold clouds without a large bias introduced by the choice of the reanalysis product. We chose 2.5 K based on previous estimates of the validity of atmospheric reanalysis temperatures in the Arctic: Jakobson et al. (2012) found a bias of up to 2°C for the lowest 890 m when comparing tethersonde data from an Arctic drifting ice station with ERA-Interim reanalysis data. Graham et al. (2019) compared ERA5 reanalysis data with radiosondes launched from two ship campaigns in the Fram Strait and found a vertically averaged absolute bias of 0.3°C. Other reanalysis products in their study also showed biases of less than 0.6°C. In addition, our own comparison of the radiosonde data from Andenes with ERA5 data yielded a bias of 1°C (see manuscript). This gives us confidence in the use of different reanalysis products for the spaceborne and ground-based retrievals of the CTT.

We also would like to state that even if we would use the same reanalysis product, a temporal collocation of the satellite overpasses with the radiosondes would never yield a similar accuracy as for the ground-based observations (due to the drifting of the radiosondes, see your comment below). At the same time, we use data from satellite overpasses within a 2°x2° box around ALOMAR, which also introduces different uncertainties due to spatial collocation with the reanalysis data. Thus, applying the same reanalysis dataset would still introduce similar uncertainties than before, which is why we still apply to ECMWF-AUX product to the satellite retrieval. We are confident that the increase of the bin size of the histogram to 2.5K drastically removes uncertainties introduced by the reanalysis product, and we thank the reviewer for suggesting this.

Jakobson, E., Vihma, T., Palo, T., Jakobson, L., Keernik, H., and Jaagus, J. (2012), Validation of atmospheric reanalyses over the central Arctic Ocean, *Geophys. Res. Lett.*, 39, L10802. https://doi.org/10.1029/2012GL051591

Graham, R. M., Hudson, S. R., & Maturilli, M. (2019). Improved performance of ERA5 in Arctic gateway relative to four global atmospheric reanalyses. *Geophysical Research Letters*, 46, 6138–6147. https://doi.org/10.1029/2019GL082781

We included the following discussion regarding temperature reanalysis products (page 6, line 168 – page 7, line 179):

"We are aware that the use of different reanalysis products for temperature retrievals introduces uncertainties. But since ECMWF-AUX has been specifically designed to provide profiles of temperature from atmospheric reanalysis interpolated on the time and location of the CloudSat/CALIPSO overpass, this makes it the first choice to use in combination with the phase retrieval from the 2B-CLDCLASS-LIDAR product. To draw conclusions about the general CTT distribution of cold clouds without a large bias introduced by the choice of the reanalysis product, we pick a bin size of 2.5 K when showing the distributions in Fig. 7. The choice of 2.5 K is based on previous estimates of the validity of atmospheric reanalysis temperatures in the Arctic: Jakobson et al. (2012) found a bias of up to 2 K for the lowest 890 m when comparing tethersonde data from an Arctic drifting ice station with ERA-Interim reanalysis data. Graham et al. (2019) compared ERA5 reanalysis data with radiosondes launched from two ship campaigns in the Fram Strait and found a vertically averaged absolute bias of 0.3 K. Other reanalysis products in their study also showed biases of less than 0.6 K. In addition, our own comparison of the radiosonde data from Andenes with ERA5 data yielded a bias of 1 K. This gives us confidence in the use of different reanalysis products for the spaceborne and ground-based retrievals of the CTT."

In addition, I doubt that indeed the cloud top temperature can be derived as accurately as it is suggested by Figure 7 a+b. Here the histogram bins have been set to 1K, which seems rather accurate when considering, e.g., that only two radiosondes were launched per day and the radiosonde may have

drifted up 20-50km (e.g., according to Seidel, JGR, 2011, https://doi.org/10.1029/2010JD014891) until it reaches the upper troposphere / lower stratosphere.

> Thanks for pointing this out. We totally agree and changed the bin resolution to 2.5 K. The choice of 2.5 K is explained in further detail in the answer to the comment above, as well as the statement included in the revised manuscript.

Besides these comments, I have some minor comments:

A general comment: Consider reducing connector words, like "however, nevertheless, …". Especially however is used rather often (e.g., 3 times between lines 37 and 45).

> Thanks for pointing this out. We reduced the usage of "however".

Page 2, line 55: Mention the difficulties of satellites to detect lower clouds due to ground clutter.

> Thanks for the comment. We added this, although our study focuses on clouds above 4km, so the effects of low cloud detection challenges are not expected to have much impact on this study. The added sentence is as follows (page 2, line 55-56):

> "Also, ground clutter can affect cloud detection, especially of low clouds."

Page 3, line 72: Define "cold cloud"

> Thanks for pointing out the first usage of 'cold cloud' without actually defining it properly. We added a short version of the definition given in chapter 4 already at this point (page 3, line 74-76):

> "[…] focus on the properties of mid- and high-level mixed-phase and cirrus clouds in this region *(single-layer clouds with cloud base heights between 4000 and 12000 m and a cloud top temperature below -20 °C)*."

Page 3, line 86: Add "profiling" before "the middle and upper atmosphere"

> This is added (page 3, line 86-87):

> "The lidar is part of the Arctic Lidar Observatory for Middle Atmosphere Research (ALOMAR) and co-located with other lidars specialized on *profiling* the middle and upper atmosphere."

Page 4, line 112: Better reword: "The cloud optical depth is another…"

> This sentence has been rephrased. In addition to a better start of the sentence we also omitted the repetition of "property" (page 4, line 114):

> "The cloud optical depth τ is another important property when considering a cloud's impact on radiative fluxes."

Page 5, line 123: Explain how you get the molecular backscatter.

> The molecular backscatter is calculated from the air density profile which again is calculated from a modified US standard atmosphere profile (adjusted to measured ground temperature and pressure). A sentence about this is added to the manuscript (page 5, line 125-127):

"[…] such that the total backscatter above the cloud is closest to the molecular backscatter. *The latter is calculated using the air density profile of a modified US standard atmosphere, i.e. adjusted to the measured ground temperature and pressure.*"

Page 5, line 132: How many false classifications were manually detected? What could cause such a false classification?

There is a reasonably high number of false detections, and this is due to calculated noise levels tending to be too low. False detections happened mostly at high altitudes above 10km where noise levels tend to be higher. Because of the described effect, the algorithm is for sure not ideal yet for automatic cloud identification, but it detects rather too many than too few cases and serves our purpose for an automated calculation of cloud base and top heights.

A sentence about the connection between false identifications and noise is added in the manuscript (page 5, line 135-138):

"*Due to the calculated noise levels tending to be too low, there is a significant number of false identifications that are actually not standing out from the background noise, especially occurring above 10 km altitude.* To avoid *these* false identifications, all clouds detected by the algorithm are manually verified."

Page 5, line 136: How far is Bodø from Andenes?

The great-circle distance is 240km. This is added in the text now (page 5, line 142-143):

"Before, the closest releases were from Bodø (67.28°N, 14.45°E) which *in a great-circle distance of 240 km* is too far away to be considered relevant for routinely retrieving CTTs."

Page 6, line 165: Remove "This increases the number of satellite overpasses"

This is done and the sentence rephrased accordingly (page 7, line 185-187):

"We include all satellite overpasses within a 2°x2° box around ALOMAR in the analysis. This corresponds to an extent of approximately 80 km in zonal and 220 km in meridional direction and results in 48873 individual profiles being used for the statistical analysis of cold cloud properties."

Page 9, line 209: Add "cirrus" before "category" and change "found" to "observed"

Thanks, this is changed (page 9, line 232):

"This is also the most common *cirrus* category *observed* at Ny-Ålesund, Svalbard, […]"

Page 9, line 223: I suggest rewording this paragraph and stating the information, that the radiosonde did not penetrate the observed layer already at the beginning.

Thanks for the comment. We changed the paragraph to the following (page 10, line 242-247):

"The radio sounding closest in time to the observation was released from Andenes at 11:04 UTC, i.e. ca. 25 min after the end of the cloud observation. Therefore, it did not penetrate the cloud, but the air mass behind it. The pronounced cloud boundary can be seen in Fig. 3a. Nevertheless,

the sounding profile reveals a temperature of -24 to -26 °C in the relevant altitude region (Fig. 3c). It increases 2 K right above 5.6 km, at the same altitude where the dew point temperature drops more than 10 K. This indicates a sudden decrease in humidity marking the border between two air masses of which the lower one is connected to the cloud and the upper one is warmer and dry above the cloud top."

Page 9, line 230: What is meant by "the center of the cloud"? I guess the base of the liquid dominated layer. Later the virga is described as "below cloud", which sounds like, the base of the liquid-dominated layer is the cloud base. Please be more specific.

Thanks for the comment. Indeed, we meant the lower part of the liquid dominated layer when writing center. This is now clarified, we give altitude references, write "below" instead of "below cloud" for the virga in the first place and use "liquid cloud base" later (page 11, line 253-259):

"The depolarization ratio in the mixed-phase cloud case shows a clear separation *into* three regions (see Fig. 4b): The center of the cloud *around 5.2-5.3 km* with values below 0.1 can be clearly separated from the cloud top and a large area below that extends down to 4.5 km (both with values up to 0.4). Thus, spherical liquid water droplets dominate in the region around 5.2 km with $\delta$ < 0.1, whereas the high depolarization values between 4.5 and 5.2 km altitude can be attributed to falling ice crystals (virga). The increasing depolarization ratio *from the liquid cloud base* towards the cloud top can be attributed to multiple scattering by liquid water droplets, since the cloud is optically thick such that there is no signal coming back from above the cloud."

Page 11, line 266: "This is less …". What does "this" refer to?

It refers to the 15.4% from the last sentence. The value is now repeated to avoid confusion (page 12, line 290):

"*This fraction of 15.4%* is less than the ratio one out of four as reported by Nomokonova et al. (2019) […]"

Page 13, Figure 6: I suggest combining Figure 6 a+b into one plot. The same holds for Fig. 7 c+d (I am not sure about Fig. 7 a+b).

Thanks for the suggestion. We combined Figures 6a+b. For 7c+d, we left them separately because putting 7a+b together would have resulted in a too busy plot and we preferred to keep the overall layout of Fig. 7 as it allows side-by-side comparison.

Page 13, line 290/291: remove "as well"

This is changed (page 13, line 314):

"The distribution from ground-based lidar has a second (lower) maximum at 212 K, […]"

Page 13, line 302/303: please change Barrow to UtqiaÄ¡vik

Thanks for the comment! "Barrow" is changed to "Utqiagvik" (page 15, line 327-328). Same also where Barrow was mentioned in the introduction (page 3, line 62-63).

Page 14, line 308: Remove "In terms of geometrical properties"

Done, the new sentence is (page 15, line 333 – page 16, line 334):

"Nakoudi et al. (2021a) also find a mean thickness of 2 km and higher cloud bases during summer and autumn than during winter and spring."

Page 16, line 329/330: "ice nucleating particle concentrations are often sufficient to completely glaciate single-layer clouds at the given temperatures". This statement is too strong, as some of the observed clouds may have formed via homogeneous ice formation.

Thanks for pointing out that this statement could be misunderstood. We definitely do not expect all the 95% to have formed via heterogeneous nucleation, by writing "often" we simply meant that heterogeneous ice nucleation must have played a non-negligible role. The sentence is rephrased as follows (page 16, line 353-355):

"This result suggests that ice formation via homogeneous freezing or, at temperatures above -38°C, heterogeneous freezing through ice nucleating particles is mostly sufficient to completely glaciate the single-layer clouds at the given temperatures."

---

## Author Comment (AC2)

RC 2 answer:

**Comments:**

The authors present ground-based lidar observations of cold clouds in Andenes (Norway), covering a period of seven years. They explored two case studies to assess 1) the agreement between a co-located cirrus observation from ground-based lidar and CALIPSO, and 2) the ground-based lidar's capability of determining cloud phase in mixed-phase clouds from depolarization measurements. Also, they presented statistics of cold clouds macrophysical properties for the period 2011-2017.

I find the manuscript interesting and well-organized. I have the following comments that require clarifications before publication of the manuscript.

> We want to thank Reviewer 2 for the helpful comments, which definitely helped to improve our manuscript. Please find our point-by-point responses to your comments below. When citing the text from the revised manuscript in the responses, we refer to page and line numbers in the new manuscript. If only single words are changed in sentences, they are given in italic font.

**Clarification needed**:

- The authors mentioned that they used 137 ERA5 pressure levels. What is known is that ERA5 are available at 137 model levels (not pressure level) and 37 pressure levels (coarse). Please clarify if some conversion procedures have been used or correction is needed?

  o Thanks for pointing out this inconsistency. We used 37 pressure levels. The 137 model levels are not relevant for the article and the number should not be mentioned. The number of pressure levels in the article is changed to 37 (see page 5, line 149-150).

- Based on Figure 1, in average ERA5 can overestimates (underestimates) cloud top temperature with a difference that can reach ~10 K. Please, elaborate on the effect of these differences on your results and conclusion, especially for the period before 2014?

  o Thanks for the comment. Considering the uncertainties, we admit that the bin width of 1 K in Fig. 7a+b of the submitted version suggests a higher accuracy than there actually is. Therefore, we decided to increase the bin size to 2.5 K, thus enabling us to draw conclusions about the general CTT distribution of cold clouds without a large bias introduced by the choice of the reanalysis product. We chose 2.5 K based on previous estimates of the validity of atmospheric reanalysis temperatures in the Arctic: Jakobson et al. (2012) found a bias of up to 2°C for the lowest 890 m when comparing tethersonde data from an Arctic drifting ice station with ERA-Interim reanalysis data. Graham et al. (2019) compared ERA5 reanalysis data with radiosondes launched from two ship campaigns in the Fram Strait and found a vertically averaged absolute bias of 0.3°C. Other reanalysis products in their study also showed biases of less than 0.6°C. In addition, our own comparison of the radiosonde data from Andenes with ERA5 data yielded a bias of 1°C (see manuscript). This gives us confidence in the use of different reanalysis products for the spaceborne and ground-based retrievals of the CTT.

Jakobson, E., Vihma, T., Palo, T., Jakobson, L., Keernik, H., and Jaagus, J. (2012), Validation of atmospheric reanalyses over the central Arctic Ocean, *Geophys. Res. Lett.*, 39, L10802, doi:10.1029/2012GL051591.

Graham, R. M., Hudson, S. R., & Maturilli, M. (2019). Improved performance of ERA5 in Arctic gateway relative to four global atmospheric reanalyses. *Geophysical Research Letters*, 46, 6138– 6147. https://doi.org/10.1029/2019GL082781

We included the following discussion regarding temperature reanalysis products (page 6, line 168 – page 7, line 179):

"We are aware that the use of different reanalysis products for temperature retrievals introduces uncertainties. However, since ECMWF-AUX has been specifically designed to provide profiles of temperature from atmospheric reanalysis interpolated on the time and location of the CloudSat/CALIPSO overpass, this makes it the first choice to use in combination with the phase retrieval from the 2B-CLDCLASS-LIDAR product. To draw conclusions about the general CTT distribution of cold clouds without a large bias introduced by the choice of the reanalysis product, we pick a bin size of 2.5 K when showing the distributions in Fig. 7. The choice of 2.5 K is based on previous estimates of the validity of atmospheric reanalysis temperatures in the Arctic: Jakobson et al. (2012) found a bias of up to 2 K for the lowest 890 m when comparing tethersonde data from an Arctic drifting ice station with ERA-Interim reanalysis data. Graham et al. (2019) compared ERA5 reanalysis data with radiosondes launched from two ship campaigns in the Fram Strait and found a vertically averaged absolute bias of 0.3 K. Other reanalysis products in their study also showed biases of less than 0.6 K. In addition, our own comparison of the radiosonde data from Andenes with ERA5 data yielded a bias of 1 K. This gives us confidence in the use of different reanalysis products for the spaceborne and ground-based retrievals of the CTT."

- In addition to the spatial difference, what is the average time lag between the radiosonde and ERA5 (for cloud top temperature)?

  - The ERA5 temperature is from the closest full hour to the measurement time. The radiosonde is only released twice a day. The average time lag between both temperature retrievals is 3 hours and 20 minutes. This is now added to the manuscript (page 5, line 150-151):

    "Additionally, the average time lag between the retrieved ERA5 temperature (available at full hours) and the radiosonde release (twice a day) in Fig. 1 is 3 hours and 20 minutes."

- Also, the vertical resolution of ERA5 at pressure levels is still coarse (not the case for model level, especially at the cirrus cloud levels). Using the interpolation can omit some important details, especially for thin cirrus. Please elaborate?

- That's right, the resolution of ERA5 pressure levels is still coarse. We added the following sentence in addition to the clarifications regarding temperature reanalysis stated above (page 5, line 145-146):

  "The rather coarse vertical resolution of the ERA5 reanalysis might omit details in the thermal structure around cirrus clouds."

- Line195: Which method is used to estimate the tropopause? Please clarify?

  - Thanks for pointing out this missing information. We apply the lapse rate definition from the World Meteorological Organization, stating that the tropopause starts where the temperature decrease with height stops. This is added in the manuscript (page 9, line 216-218):

    "*Applying the lapse rate definition of the tropopause by the World Meteorological Organization (WMO)* we estimate the *beginning of the* tropopause to be located at about 11.0 km (from radiosonde) or 10.6 km (from reanalysis data) *and* at a temperature of -70 °C."

- Please clarify further about phase discrimination between cirrus, mixed-phase and liquid clouds, during maintenance break from April 2013 to July 2015?

  - We did not include the time period April 2013 to July 2015 for the ground-based observations, as the maintenance break made measurements impossible. Thus, also no phase discrimination has been applied for this period. The phase discrimination for the 2B-CLDCLASS-LIDAR product is explained in Sassen et al. (2008) and referenced in Sect. 2.2. (No change in the manuscript.)

- The authors mentioned that "…. Thus, the cirrus cloud is extending well into the tropopause, dehumidifying the upper troposphere and lower stratosphere region through ice crystal growth and sedimentation."

- Can you provide evidence (quantification) on dehumidifying the lower stratosphere caused by cirrus? Also, on cirrus reaching lower stratosphere causing dehydration.

- I would like to see figures and discussion about the corresponding relative humidity with respect to ice (in-cloud and clear-sky) associated with cirrus cloud. Also, its impact on cirrus cloud, dehydration, and your conclusion.

  - For this specific case, from elastic lidar data only, we can't quantify the amount of dehydration. But we can see that the relative humidity with respect to ice is significantly larger than 100% inside the cloud and until 11.7km, which corresponds roughly to the cloud top and is above the beginning of the tropopause. For the sake of clarity in presentation we did not add an extra figure showing the relative humidity with respect to ice, but for demonstration we added the frost point temperature to the following sounding plot.

[Figure]

**2011-04-01 11:10 UTC**
*Bodø*: 67.29 °N 14.45 °E

There we can see that the decrease in frost point temperature and thereby humidity first begins well above the altitude where the temperature decrease stops. Therefore we can conclude that the cirrus cloud removes vapor from the tropopause by forming ice crystals that then fall and evaporate in the troposphere. But again, quantitative studies of dehydration are not possible with this setup and beyond the scope of the study. The case is meant as a qualitative example.

To make limitations clearer, we added the word "potentially" before "dehumidifying" in the manuscript and added one extra sentence. The new text is (page 9, line 218-221):

"Thus, the cirrus cloud is extending well into the tropopause, *potentially* dehumidifying the upper troposphere and lower stratosphere region through ice crystal growth and sedimentation (e.g., Kärcher, 2005). *However, a quantification of dehydration in this case requires knowledge of further cloud parameters and is beyond the scope of this study.*"

**Minor comments**:

Correct "occurence" --> occurrence. (lines 8, 22, 57, 64,66, 291 and y-axis of Fig.5b)

Thanks for pointing that out. It is now changed in both text and figures.